# The Role of Tryptophan Metabolites in Musculoskeletal Stem Cell Aging

**DOI:** 10.3390/ijms21186670

**Published:** 2020-09-11

**Authors:** Jordan Marcano Anaya, Wendy B. Bollag, Mark W. Hamrick, Carlos M. Isales

**Affiliations:** 1Universidad Central Del Caribe Laurel, Av. Sta. Juanita, Bayamón PR 00960, Puerto Rico; 119jmarcano@uccaribe.edu; 2Department of Physiology, Augusta University and Charlie Norwood VA Medical Center, Augusta, GA 30912, USA; wbollag@augusta.edu; 3Department of Cellular Biology and Anatomy, Augusta University, Augusta, GA 30912, USA; mhamrick@augusta.edu; 4Departments of Medicine, Neuroscience and Regenerative Medicine, Augusta University, Augusta, GA 30912, USA

**Keywords:** stem cells, kynurenine, tryptophan metabolites

## Abstract

Although aging is considered a normal process, there are cellular and molecular changes that occur with aging that may be detrimental to health. Osteoporosis is one of the most common age-related degenerative diseases, and its progression correlates with aging and decreased capacity for stem cell differentiation and proliferation in both men and women. Tryptophan metabolism through the kynurenine pathway appears to be a key factor in promoting bone-aging phenotypes, promoting bone breakdown and interfering with stem cell function and osteogenesis; however, little data is available on the impact of tryptophan metabolites downstream of kynurenine. Here we review available data on the impact of these tryptophan breakdown products on the body in general and, when available, the existing evidence of their impact on bone. A number of tryptophan metabolites (e.g., 3-hydroxykynurenine (3HKYN), kynurenic acid (KYNA) and anthranilic acid (AA)) have a detrimental effect on bone, decreasing bone mineral density (BMD) and increasing fracture risk. Other metabolites (e.g., 3-hydroxyAA, xanthurenic acid (XA), picolinic acid (PIA), quinolinic acid (QA), and NAD+) promote an increase in bone mineral density and are associated with lower fracture risk. Furthermore, the effects of other tryptophan breakdown products (e.g., serotonin) are complex, with either anabolic or catabolic actions on bone depending on their source. The mechanisms involved in the cellular actions of these tryptophan metabolites on bone are not yet fully known and will require further research as they are potential therapeutic targets. The current review is meant as a brief overview of existing English language literature on tryptophan and its metabolites and their effects on stem cells and musculoskeletal systems. The search terms used for a Medline database search were: kynurenine, mesenchymal stem cells, bone loss, tryptophan metabolism, aging, and oxidative stress.

## 1. Introduction

The size of the aging population (65 years or more) is projected to increase by about 60% by 2030 [1]. This poses a challenge to the healthcare system because of the associated increase in age-related chronic diseases. There is an urgent need for new treatments to address these issues, requiring an increase in research on aging and the aging process. Dementia, cardiovascular diseases, frailty, sarcopenia, and osteoporosis are examples of different conditions that significantly impact the lives of this population and for whom age is a major risk factor. Specifically, osteoporosis was estimated to affect almost 54 million Americans aged 50 years or older in 2010 [2]. Osteoporosis is characterized by a decrease in bone mineral density (BMD) and structural deterioration of the bone. The frequently associated condition of physical frailty predisposes falling and the combination of frailty and osteoporosis results in an increased risk of fractures [3]. Treatment for bone fractures represent an economic burden for patients as the average incremental direct cost increase in the 6 months following a long bone fracture ranged from $5707 to $39,041 in 2012 [4].

Mesenchymal stem cells (MSCs) are multipotent progenitor cells mainly found in the bone marrow (BM) that can differentiate into osteoblasts and thus promote bone formation [5]. Hypoxic conditions within healthy BM are thought to influence the proliferation and cell-fate commitment of hematopoietic stem cells (HSCs) and MSCs [6,7,8]. However, with age, there is a marked increase in oxidative stress within the BM and a loss of the normal microarchitectural niches because of osteoporosis, which alters the hypoxic environment and, thus, impairs normal stem cell function [9]. This increase in reactive oxygen species (ROS) and its associated damage to MSCs contributes to the pathogenesis of many age-related diseases. An increase in ROS has been associated with decreased Wnt, Hedgehog, BMP, and ERK signaling pathways, all of which are important for bone formation [10]. Furthermore, an impairment of MSC differentiation into osteoblasts has been found in patients with osteoporosis due to an imbalance of bone deposition and resorption [11]. The specific pro-oxidative agents that cause such downregulation remain unclear and represent an area for further investigation.

Aromatic amino acids (tryptophan, phenylalanine, and tyrosine) are generally known to have antioxidant properties and to play a role in MSC survival and proliferation. Tryptophan (Trp) in particular has been shown to upregulate ERK phosphorylation/activation in bone marrow stromal cells (BMSCs); ERK activation is involved in the cell response to extracellular proliferation signals [12]. Additionally, Trp was found to upregulate the Akt pathway and FOXM1 of an aging BM under normoxic conditions (21% O_2_). The Akt pathway promotes cell cycle progression and proliferation, whereas FOXM1 is a transcription factor known to protect cells against oxidative stress [13]. Trp metabolism through the kynurenine pathway has also been associated with modulation of age-related chronic inflammation, known as inflammaging [14]. As a result, the kynurenine (Kyn)/Trp ratio has been used as a biomarker to detect inflammaging and the onset of age-related diseases. With age, this ratio has been reported to increase, either due to an age-dependent decrease in Trp levels and/or an increase in Kyn [14,15]. Evidence suggests that elevated levels of Kyn with age are linked to a higher incidence of chronic diseases and with a lower life expectancy in several adult mammals [16,17]. In this review, we will discuss in detail different Trp metabolites with a particular focus on those of the Kyn pathway (Figure 1). Specifically, we will focus on what is the known about the impact of Kyn pathway metabolites, including serotonin and melatonin, on bone. Although the kynurenine pathway has been previously studied in relation to CNS abnormalities, its role in musculoskeletal disorders is an emerging area of research. The current review is meant as a brief overview of existing English language literature on tryptophan and its metabolites and their effects on stem cells and musculoskeletal systems. The search terms used for a Medline database search were: kynurenine metabolites, mesenchymal stem cells, bone loss, tryptophan metabolism, aging, and oxidative stress and although the major emphasis was on more recent articles (2010–2020), the search extended back to 1984.

## 2. Kynurenine Impacts MSCs and Causes Age-Related Bone Loss

We have previously shown that with aging, both MSC number and differentiation capacity decrease. The mechanism responsible for these changes is not known [18]. Kyn is produced from Trp breakdown through either tryptophan 2,3-dioxygenase (TDO), mostly in the liver, or via indoleamine 2,3-dioxygenase (IDO) −1 or −2, occurring in most tissues [19]. Specifically, IDO is readily activated by pro-inflammatory cytokines, with interferon-gamma (IFN-γ) being the most potent [20,21]. IDO-dependent activation of the Kyn metabolic pathway has been found to inhibit osteogenesis and bone formation [19]. This enzyme is expressed in macrophages and to a lesser extent in MSCs [19]. As osteoclasts are derived from macrophages, one could infer that they are the principal IDO-expressing cells in bone and play a major role in Kyn production. Within bone, Kyn has been associated with multiple age-related phenotypes seen in patients with osteoporosis. In a study by El Refaey et al. [19], feeding 12-month-old mice increasing Kyn concentrations (50 μM and 100 μM) mimicked the bone aging process resulting in bone parameters comparable to those of a 24-month-old mouse. These authors reported lower bone volume (BV), lower bone volume over total volume (BV/TV), increased number of TRAP-labeled osteoclasts, greater bone resorption marker levels, and decreased Hdac3 expression, with Kyn feeding [19]. Such parameters are similar to those observed in aged bone. Reduction in Hdac3 expression was linked to higher BM adiposity, which is also seen with aging [22].

Recently, a number of studies have focused on the potential mechanisms through which Kyn could induce an aging phenotype in bone. These include: enhanced MSC senescence, decreased autophagy, increased oxidative stress, reduced osteogenic factors, and altered miRNA expression, among others [23,24,25]. Although much of the bone-related research focuses on Kyn, not much is known about other Trp metabolites, specifically those downstream of Kyn. Further research in this area would help in targeting specific pathways in Trp metabolism for pharmacological treatment of bone loss.

### 2.1. Kynurenic Acid Antagonizes NMDA Receptors Found on Osteoclasts

Kynurenine can be further catabolized into various downstream products such as kynurenic acid (KYNA). This metabolite is well-known for its neuroprotective role in the central nervous system (CNS) because of its antagonism of glutamate receptors such as the NMDA, AMPA, and kainate receptors [26]. Thus, increased KYNA levels are associated with amelioration of the excitotoxic conditions seen in age-related disorders like Alzheimer’s (AD) and Parkinson’s disease. Moroni et al. [27] found that there was an increase in the KYNA content in the brain and blood of 18-month-old rats, compared to 3-month-old rats. However, KYNA levels failed to increase in the liver and kidneys of aged rats [27]. This suggests an organ-specific increase in KYNA with age. On the other hand, the effects of KYNA on bone are not completely clear. One study confirmed the presence of ionotropic glutamate receptors in osteoblasts, osteocytes, and osteoclasts, the latter having the highest concentration. They also found that blocking the NMDA receptor in osteoclasts inhibited bone resorption [28]. These data would suggest that KYNA should antagonize the resorptive action of osteoclasts in bone. Our group conducted a study to explore the effects of this metabolite on bone and found that treatment with KYNA resulted in bone loss in the periphery [29], although the mechanism responsible for this bone loss is still to be elucidated. KYNA effects on MSCs, specifically, are not known.

### 2.2. 3-Hydroxykynurenine Increases Oxidative Stress and Leads to Cytotoxic Effects

Via the enzyme Kyn monooxygenase, 3-Hydroxykynurenine (3HKYN) is produced by hydroxylation of Kyn. In contrast to KYNA, 3HKYN is mostly known for its cytotoxicity through its ability to induce ROS leading to apoptosis [30,31]. This oxidation product has been associated with several age-related diseases such as Huntington’s disease (HD) and cataracts [32,33]. Regarding bone aging phenotypes, oxidative stress has been found to reduce osteogenic markers in MSCs [10]. In a study conducted by Fakondut et al. [33], the capacity of 3HKYN to reduce osteoblast-like cell (MC3T3-E1 cells) viability due to its pro-oxidative nature was tested. These authors found that at doses of 250 μM to 1 mM over a period of 60 h, 3HKYN reduced cell viability and further than blocking its effects on oxidative stress resulted in restored cell viability [34]. However, the 3HKYN concentrations used were much higher than those seen physiologically, and concentrations of 3HKYN that are usually seen with aging were not studied. Nevertheless, in humans 3HKYN was associated with an increased risk of hip fractures in elderly women over a 9 to 11-year period [35]. Further studies should address the specific effects of 3HKYN within aging cells and whether its concentration increases in bone with aging.

### 2.3. The 3-Hydroxyanthranilic Acid to Anthranilic Acid Ratio

Anthranilic acid (AA) is also part of the Kyn pathway and can be directly produced from Kyn through the enzyme kynureninase. Not much is known about direct AA effects on the body, but synthetic derivatives have been used as immunosuppressive and anti-inflammatory drugs [36,37]. 3-Hydroxyanthranilic acid (3HAA) can be a product of either 3HKYN or AA. This metabolite is reported to have both antioxidant and pro-oxidant behavior depending on its chemical environment [38]. When in a pro-oxidant environment, 3HAA auto-oxidizes by dimerizing into cinnabarinic acid [39]. 3HAA has been found to prevent β-amyloid aggregation by binding to a specific region within the protein that, in the absence of 3HAA, promotes its misfolding [40]. This represents a potential research area to target for Alzheimer’s disease (AD). A study by Darlington et al. [41] assessed the importance of the 3HAA:AA ratio as a biological marker to indicate the progression of disorders with an inflammatory component. They found that this ratio was significantly reversed in patients with osteoporosis, as patients showed not only lower baseline levels for 3HAA, but also higher levels of AA compared to control [41]. Some studies suggest that a possible explanation is an inhibition of the enzymatic conversion of AA to 3HAA, thus resulting in lower levels of 3HAA [42]. Another project from the Hordaland Health Study examined the relationship between different Kyn pathway metabolites and BMD. They found a positive correlation between BMD and 3HAA in differently aged groups of both men and women [43]. Nevertheless, much more knowledge about these metabolites is needed to understand their impact on aging bone and MSCs.

## 3. Xanthurenic Acid Induces Cell Apoptosis

Xanthurenic acid (XA) is also produced from 3HKYN by an enzymatic reaction catalyzed by Kyn aminotransferase. XA is a well-studied Trp metabolite that is known to trigger apoptosis by promoting the release of cytochrome C, resulting in the destruction of mitochondria as well as structural proteins such as gelsolin [44]. It is also reported that XA induces the translocation of several pro-apoptotic Bcl-2 family proteins to mitochondria, thus inducing apoptotic mechanisms [45]. This metabolite accumulates in aging tissues and has been associated with certain degenerative diseases, such as senile cataracts, through its ability to promote cell death and disrupt the normal physiology of the lens epithelium [46]. Much data exist about the cytotoxic impact of XA and its negative effect on the human lens, but little is known concerning its impact on aging bone. Considering the data obtained, one could hypothesize that XA would likely negatively affect bone and lead to an aging phenotype. However, the same Hordaland Health Study project mentioned earlier found a positive correlation between BMD and XA [43]. These results suggest a novel mechanism of XA effects on aging bone.

### 3.1. Picolinic Acid Increases Bone Marrow Adiposity

Picolinic acid (PIA) results from the conversion of 3HAA to 2-amino-3-carboxymuconate-6-semialdehyde (ACMS) and then to PIA by an enzymatic reaction. This metabolite is mostly known as a metal chelator, and thus, PIA has been used as a way to introduce chelators into biological systems [47]. Some studies report PIA as having a neuroprotective role because it has been shown to block the neurotoxic effects of quinolinic acid (QA) within the cortex of the brain [48,49]. However, levels of PIA within the CNS in the context of neurodegenerative diseases have yet to be explored. Regarding bone, Vidal et al. [50] conducted a series of tests to explore the effects of PIA. They found that IDO knockout murine MSCs that were treated with increasing concentrations of PIA recovered their osteogenic potential. They also reported that PIA-treated human MSCs (hMSCs) increased their expression of osteogenic genes like Runx2 and osteocalcin (OCN) in vitro [50]. Our group conducted a study to test the in vivo effects of PIA in mice. We found that feeding PIA did not impact BMD, trabecular bone, or bone microstructure, but did result in increased BM adiposity. Furthermore, lipid storage genes such as Plin1 and Cidec were increased in PIA-treated MSCs [51]. Notably, these two studies not only were performed in different systems (in vitro versus in vivo) but also used different concentrations of PIA, indicating that there might be a dose dependence to the effect of this metabolite on bone.

### 3.2. Quinolinic Acid Is a Neurotoxin Related to Neurodegenerative Diseases

Quinolinic acid (QA) is formed from ACMS through a non-enzymatic reaction. In contrast to PIA, this metabolite is widely known for its excitotoxicity via its activation of NMDA receptors [52]. In a study by de Bie et al. [53], the cerebrospinal fluid of 49 women of all ages (0–90 years) was tested and a positive correlation between QA and age determined [53]. Additionally, concentrations in the cortex of mammals like rats were reported to increase with advancing age [54]. As a result, QA has been implicated in multiple neurodegenerative diseases including AD, HD, and amyotrophic lateral sclerosis (ALS) [55]. Outside the CNS, not much is known about the impact of QA on the body; however, one might expect QA to promote bone loss through its activation of the NMDA receptor. In the same Vidal et al. [50] study mentioned earlier, the impact of QA on ex vivo MSCs lacking IDO was tested. Contrary to expectations, QA increased the osteogenic potential of these cells, but to a significantly lesser extent than PIA [50]. Like picolinic acid, the impact of QA on MSCs needs to be addressed in an in vivo setting.

### 3.3. Nicotinamide Adenine Dinucleotide (NAD+) Decline Is Related to the Appearance of Age-Related Diseases

Nicotinamide adenine dinucleotide (NAD+) is the terminal oxidation product of Trp breakdown through the Kyn pathway and is a key cofactor in multiple biological processes. Although its better-known function is its role in mitochondrial redox reactions and ATP production, it also plays an important role in anabolic pathways for many macro-nutrients [56]. NAD+ is known to decline with aging and this decline coincides with the appearance of age-related diseases [56,57,58]. Even though some groups suggest a depletion of Trp with age, Schultz & Sinclair report that CD38, a NADase, increases in certain tissues with age and its inhibition leads to increased NAD+ [57], suggesting that changes in NAD+ may play an important role in aging pathology. This decline in NAD+ levels is reported to cause mitochondrial dysfunction and an inability to repair DNA damage. It is also associated with diabetes, atherosclerosis, and AD [59]. A study by Iqbal & Zaidi assessed the role of NAD+ and its metabolites (ADP-ribose and cyclic ADP-ribose) in osteoclastogenesis. Although their results showed no direct impact of NAD+, they found that increasing concentrations of cyclic ADP-ribose had an osteoclastogenic effect, as did the addition of exogenous ADP-ribosyl cyclase. On the other hand, its non-cyclic form had an inhibitory effect on osteoclast formation [60].

These data have suggested a potential therapeutic role for NAD+ supplementation to prevent age-related decline in function. A review presented by Aman et al. [60] reported that long-term administration of NAD+ precursors such as nicotinamide mononucleotide (NMN) was able to counter some hallmarks of aging and increase insulin sensitivity and bone density and improve immune function [61]. The impact of NAD+ repletion on muscle stem cell (MuSC) function in aged mice was also examined; investigators found that treatment with nicotinamide riboside (NR), an NAD+ precursor, increased the number of MuSCs and enhanced muscle function [62].

### 3.4. The Dual Role of Serotonin in Bone Homeostasis

In contrast to the Kyn pathway, much more is known about the impact of serotonin (SE) on bone and organ function. SE is derived from Trp through a two-step reaction in which tryptophan hydroxylase (Tph) 1 or 2 is the rate-limiting enzyme. Tph1 is mainly found in the pineal gland and the gut, whereas Tph2 is the main enzyme in serotonergic neurons [63,64]. This molecule is widely known for its role as a neurotransmitter in the CNS where it is derived mainly from the raphe nuclei in the brainstem. There are multiple responses to SE known depending on the receptor expressed by the cell; however, it is mostly recognized for its role in preventing depressive and anxiety-like symptoms. Brain-derived serotonin (BDS) levels have been found to decrease with age and to be associated with late-life depression and AD [65]. On the other hand, most SE is found in the periphery where it is produced mainly in the gut. Gut-derived serotonin (GDS) is known to regulate gut motility, platelet aggregation, and bone homeostasis [66]. Nonetheless, not much is known about how GDS changes with age and the impact of gut microbiota on its levels.

The effects of SE on bone are known to be complex, as GDS and BDS have opposite effects on the skeletal system. Mice lacking Tph2 show a bone loss phenotype, with increased bone resorption and lower BV/TV [67]. This result suggests that BDS plays a key role in bone homeostasis. SE can also promote bone formation by decreasing sympathetic tone through Htr2c receptors found in the ventromedial nucleus in the hypothalamus (VMH) [67]. Increased sympathetic tone is known to promote bone loss by the interaction of leptin with ObR receptors also found in the VMH [68,69]. This interaction in the bone occurs through β2-adrenergic receptors found on osteoblasts and leads to mechanisms that reduce osteoblast proliferation and increase osteoclast differentiation [69]. GDS is known to increase bone resorption and to negatively impact the skeletal system. SE produced in the periphery does not cross the blood-brain barrier into the CNS.

Abnormalities in the tryptophan/serotonin/kynurenine pathway have also been implicated in the pathogenesis of the bone disease associated with chronic kidney disease (CKD-MBD) [70,71]. Kalaska et al. [72] examined the correlation between serum kynurenine metabolites and parameters of bone turnover in rats after partial nephrectomy. They found that after one month, the bones of nephrectomized rats had a significant increase in osteoclast number as well as decreased cortical bone mineral density. These changes were associated with a significant increase in serum levels of both KYN and 3-HKYN. There was also significant upregulation of expression of the AhR gene, presumably the mediator of KYN effects. In contrast to peripheral effects, central kynurenine metabolism (similarly to serotonin) may have a beneficial impact on bone parameters in the setting of renal insufficiency [73]. Rats undergoing partial nephrectomy were found to have a significant elevation in KYN with a decrease in tryptophan levels in brain areas examined (cerebellum, brainstem, frontal cortex, hypothalamus and striatum) and an increase in tibial cross-sectional area and wall thickness.

Lrp5-deficient mice have increased Tph1 [74,75,76] and in humans, mutation in the Lrp5 gene causes osteoporosis pseudoglioma, which is characterized by bone loss and blindness [74]. While Lrp5 does not directly promote bone loss, its regulation of GDS levels impacts bone homeostasis in patients with osteoporosis pseudoglioma [77]. A study by Yadav & Ducy explored the mechanism through which SE acts on bone and found that the Htr1b receptor was highly expressed on osteoblasts. Knockdown of this receptor resulted in a higher bone formation rate and an increased number of osteoblasts. Furthermore, these authors reported that GDS acted by decreasing cAMP response element-binding protein (CREB) and cyclin D1 interaction [75]. Another study found that high circulating levels of GDS suppressed osteoblast proliferation mechanisms by an ability to decrease association between CREB and Forkhead box protein 1 (FOXO1) transcription factors [78].

### 3.5. The Use of Melatonin as an Osteogenic Therapy

Melatonin is another Trp breakdown product and is synthesized in the pineal gland downstream of serotonin production. Its secretion is controlled mainly by the suprachiasmatic nucleus, and it has a role in regulating the sleep–wake cycle and modulating body temperature [79]. Melatonin can be produced by almost every cell in the body and is a powerful mitochondrial anti-oxidant. In fact, the gut appears to have the highest melatonin concentrations in the body (four hundred times higher than the pineal gland). Melatonin is synthetized in the enterochromaffin cells in the gut and seems to serve a paracrine function [80].

Melatonin has been reported to decrease during aging, and this reduction is associated with a deregulation of the circadian rhythms of the body and an increase in oxidative stress [81]. Recently, the therapeutic role of melatonin in preventing age-related characteristics was explored. Fang et al. [81] examined the effects of melatonin supplementation on canine adipose-derived mesenchymal stem cells. They found that melatonin attenuated cell senescence and reduced endoplasmic reticulum stress (ERS), both of which are characteristics of aging [82]. Another study explored the effect of treating murine MSCs with melatonin after an ischemic injury, which leads to increased cellular levels of ROS. In this study, melatonin decreased autophagy-mediated apoptosis and ERS by increasing prion protein PrP^C^, which is involved in reducing oxidative stress [83].

A number of recent studies have used melatonin to treat osteoporosis. Kotlarczyk et al. [84] found that melatonin treatment decreased the bone resorption to bone formation ratio (NTX:OCN) in perimenopausal women, as measured by the serum levels of the respective markers [84]. Another study showed that treating perimenopausal and postmenopausal women with melatonin resulted in increased BMD and osteocalcin, an osteogenic marker [85]. Other studies have focused on the mechanisms through which melatonin may exert its bone-forming effects. Zhang et al. [86] explored the osteogenic mechanism of melatonin on human mesenchymal stem cells (hMSCs) and discovered that it promoted the expression of Runx2. In addition, they found that it suppressed the adipogenic marker, PPAR-γ [86]. Another group reported that melatonin exerted an osteoblastogenic effect through the MT2 receptor by increasing expression of proliferation markers such as Runx2, Bmp2, and Bglap [87]. Thus, melatonin may represent a therapeutic option in the treatment of osteoporosis.

## 4. Summary

Tryptophan is an important amino acid in the body’s normal physiology, and deregulation of its metabolism has been implicated in the aging process. Although much attention has been given to the effects of kynurenine in promoting aging bone phenotypes, few studies have investigated the impact of its downstream metabolites. Data from studies presented in this review and summarized in Table 1, suggest a complex role for the different kynurenine metabolites in normal/abnormal bone physiology. For example, KYNA, 3HKYN, and AA are all associated with osteoporosis, with its decreased BMD and increased risk of fractures. In contrast, some investigators have found that 3HAA, XA, PIA, QA, and NAD+ have a beneficial impact on bone mass.

## 5. Discussion

The current review discusses available English language literature on what is known about the impact of various tryptophan metabolites on stem cells and musculoskeletal tissues. The review is limited by the fact that little is known about some of the specific metabolite effects on bone and muscle. In these cases, specific metabolite effects on other tissues are discussed.

These studies highlight the need for further research on the impact of these tryptophan metabolites on aging bone. Defining the underlying mechanisms may permit the development of therapeutics targeting specific metabolites in the Trp metabolic pathway.

Melatonin is a powerful mitochondrial anti-oxidant and could have potential therapeutic benefits in modulating the progression of age-related diseases such as osteoporosis and sarcopenia. It is also possible that it might be more practical to inhibit the kynurenine pathway and selectively add back metabolites that are of benefit; for example, Castro-Portugez and Suthphin [84] have suggested inhibition of the kynurenine pathway with selective replacement of NAD+. NAD+ is important in mitochondrial bioenergetics and supplementation has been associated with increased longevity [88]. Inhibition of the first step in this pathway by inhibiting indoleamine 2,3-dioxygenase (IDO) is already an experimental strategy that has been used for treating certain types of cancer [89]. Thus, modulation of the tryptophan catabolic pathway has the potential to benefit not only musculoskeletal aging but also other age-related diseases, such as neurodegenerative disorders, cancer, and vascular diseases.

These goals may be further complicated by the fact that the tryptophan metabolites have both endocrine and paracrine roles. For example, tryptophan breakdown occurs predominantly through the TDO pathway in the liver and in fact the majority of the kynurenine in the brain is derived from this circulating kynurenine transported through the blood brain barrier. In contrast to the setting of tissue inflammation, kynurenine is produced locally through activation of IDO. Tissue-targeted modulation of the activity of these metabolites may present a major challenge in view of the need to minimize undesirable side effects to allow therapeutic use of the various metabolites of the Kyn pathway.

## Figures and Tables

**Figure 1 ijms-21-06670-f001:**
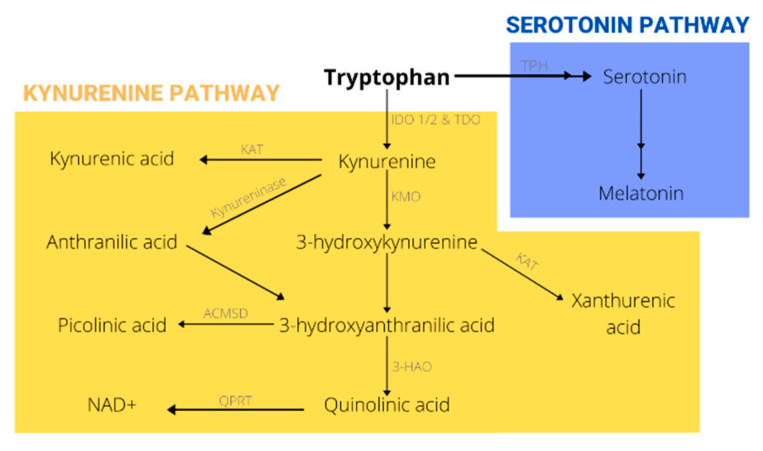
Tryptophan metabolism.

**Table 1 ijms-21-06670-t001:** Impact of tryptophan metabolites on bone characteristics.

Tryptophan Metabolites	Oxidative Stress	BM Adiposity	BMD	BV/TV	Osteogenesis	Osteoclastogenesis
1. Kyn	🠕	🠕	🠗	🠗	🠗	🠕
2. KYNA				🠗		
3. 3HKYN	🠕				🠗	
5. 3HAA			🠕			
6. XA			🠕			
7. PIA		🠕			🠕	
8. QA					🠕	
9. NAD+	🠗		🠕			*
10. GDS			🠗		🠗	🠕
11. BDS			🠕		🠕	🠗
12. ME	🠗	🠗			🠕	

Summary of the impact of tryptophan metabolism on bone homeostasis. * ADP-ribose inhibits osteoclastogenesis; cyclic ADP-ribose promotes osteoclastogenesis.

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
