# Peer review of "The Role of Tryptophan Metabolites in Musculoskeletal Stem Cell Aging"

_ijms, 2020, doi:10.3390/ijms21186670_

Round 1

Reviewer 1 Report

Thank you for the opportunity to review your manuscript.

This manuscript entitled “The Role of Tryptophan Metabolites in Musculoskeletal Aging” reports on a narrative review of the impact of tryptophan metabolites downstream of kynurenine with the emphasis on their impact on bone. The authors found that different kynurenine metabolites play different roles in bone physiology, some associated with osteoporosis and some beneficially impacting bone mass. 

Overall the manuscript is well written. It details the information in clear and concise language. The manuscript’s aim is clearly stated. The sections of the manuscript are clearly marked with distinct numbered headings.  The manuscript’s format is similar to others previously published in the international Journal of Molecular Science.

The structure was clear and obvious, however it did not include information on the strategy used to identify the information or give a clear discussion section evaluating the information.  Of concern, the manuscript does not present the methodology used to identify the articles and information included. The authors likewise do not give a quality assessment of their information provided. These pieces are essential for judging the quality and completeness of the information.

My specific requests for change follow:

1. Although the International Journal of Molecular Science does not require a methods section or discussion section for reviews, the manuscript could benefit from a description of the methods used to identify the source works included in the review.

Please consider including a methods section on how the information was identified. 

2. If you add a methods section, Sections 2-11 should comprise the results and be renumbered.

3. Even if the authors choose not to include a methods section, it would still be possible to add a discussion section. Currently, the manuscript is lacking a clear discussion section.  The second paragraph of the summary section would be well suited to be included in a discussion section, especially as it cites literature not mentioned in the previous sections.  The discussion section should include a description of the quality and completeness of the information presented and a paragraph on the limitations of the reporting. While the journal does require that authors of systematic reviews report their work following PRISMA guidelines, the requirements for reviews do not.  However, I suggest you look at the PRISMA guidelines in structuring your discussion section.

Please consider including a discussion section and report either limitations or information on the quality and completeness of the information presented.

Please include information on the limitations and or on the quality and completeness of the information presented in the summary section, if a discussion section is not created.

Minor comment:

The format of the term “tryptophan 2,3 dioxygenase inhibitor” and “indoleamine 2,3 dioxygenase” are usually written with dashes as “tryptophan 2,3-dioxygenase inhibitor” and “indoleamine 2,3-dioxygenase”, respectively.

Author Response

We thank the reviewer for their helpful and positive comments about the review. We respond to the specific issues below:

(1)  Although the International Journal of Molecular Science does not require a methods section or discussion section for reviews, the manuscript could benefit from a description of the methods used to identify the source works included in the review. Please consider including a methods section on how the information was identified. 

We have decided to add information about the review in the initial section from Lines 71-79.

2. If you add a methods section, Sections 2-11 should comprise the results and be renumbered.

Instead of a Methods section we have added this requested information to the initial section of the manuscript

3. Even if the authors choose not to include a methods section, it would still be possible to add a discussion section. Currently, the manuscript is lacking a clear discussion section.  The second paragraph of the summary section would be well suited to be included in a discussion section, especially as it cites literature not mentioned in the previous sections.  The discussion section should include a description of the quality and completeness of the information presented and a paragraph on the limitations of the reporting.

A Discussion section has been added.

Minor comment:

The format of the term “tryptophan 2,3 dioxygenase inhibitor” and “indoleamine 2,3 dioxygenase” are usually written with dashes as “tryptophan 2,3-dioxygenase inhibitor” and “indoleamine 2,3-dioxygenase”, respectively.

This has been corrected, thanks

Reviewer 2 Report

The present review tried to collect the available data on the impact of the tryptophan breakdown products, like KYN and its metabolites, serotonin and melatonin on the body  in general and, when available, the existing evidence of their impact on musculosketetal system, especially during aging process. The manuscript is well written, however, some recent data concerning the significance of serotonin and kynurenine in bone health has been omitted. Namely, the chronic kidney disease (CKD) with a concomitant renal osteodystrophy, now called as CKD-MBD (chronic kidney disease-mineral and bone disorders) is a specific illness, which resembles aging-related osteoporosis. The recent studies of Pawlak et al. performed on rat CKD model, documented the role both peripheral (Pawlak et al. PLoS One 2016 Oct 6;11(10):e0163526; Bone 2018 Aug;113:124-136; Expert Opin Ther Targets. 2019 Apr;23(4):353-364; Biochim Biophys Acta Mol Basis Dis. 2019 Nov 1;1865(11):165528) as well as brain serotonin (Pawlak et al. Bone. 2017 Dec;105:1-10) in bone serotonin-dependent molecular pathways, bone metabolism, mineral status and strength. The same team also documented the significance of peripheral and central kynurenine in bone health in this model (Kalaska et al. Front Physiol. 2017 Oct 31;8:836; PeerJ. 2017 Apr 20;5:e3199). Still now, the mechanisms involved in the cellular actions of these tryptophan metabolites on bone  are only weakly known, thus the including these new studies in the present manuscript would significantly increase its scientific value.

The minor concern: Table 1. It is well known that GDS caused bone loss, whereas in this table GDS is related to increased BMD?

Author Response

(1) Including these new studies in the present manuscript would significantly increase its scientific value.

We thank the Reviewer for pointing out these references to us and have now added a paragraph discussing these publications (Lines 251-263)

(2) The minor concern: Table 1. It is well known that GDS caused bone loss, whereas in this table GDS is related to increased BMD?

We have added up and down arrows for BMD for GDS and BDS respectively.